# Adapting to Changing Climate: Understanding Coastal Rural Residents' Relocation Intention in Response to Sea Level Rise

**Richard Adade** [1,2,]*, **Dukiya Jaiye** [3], **Nana Ama Browne Klutse** [4] **and Appollonia Aimiosino Okhimamhe** [1,5]

1   School of Physical Science, WASCAL Doctoral Research Programme in Climate Change and Human Habitat, Federal University of Technology Minna, Minna P.M.B. 65, Niger, Nigeria
2   Centre for Coastal Management, African Centre of Excellence in Coastal Resilience, University of Cape Coast, Cape Coast P.M.B., Ghana
3   Department of Urban and Regional Planning, Federal University of Technology Minna, Minna P.M.B. 65, Niger, Nigeria
4   Department of Physics, University of Ghana, Accra 00233, Ghana; nklutse@ug.edu.gh
5   Department of Geography, Federal University of Technology Minna, Minna P.M.B. 65, Niger, Nigeria
*   Correspondence: richard.adade@ucc.edu.gh; Tel.: +233-242-530-909

**Abstract:** Ex situ adaptation in the form of relocation has become inevitable in some low-lying coastal zones where other adaptation strategies become impractical or uneconomical. Although relocation of coastal low-lying communities is anticipated globally, little is still known about the factors that influence household-level adoption. This study draws on an extended version of Protection Motivation Theory (PMT) to assess the factors influencing the relocation intention of three highly vulnerable coastal rural communities in Ghana. A total of 359 household heads were randomly selected for a questionnaire survey. The study employed binary logistic regression to identify key factors that influence residents' readiness to relocate. The results indicated that cognitive and compositional factors were more important than contextual factors in explaining the intention to relocate among coastal rural communities in Ghana. However, contextual factors mediated or attenuated the influence of cognitive and compositional factors on relocation intention. Based on the findings, this study advocates for intensive education on the effects of future sea-level rise impacts on communities as well as structural and non-structural measures to improve the socio-economic capacity of rural communities.

**Keywords:** Ghana; sea level rise; coastal rural community; climate change adaptation; relocation; perceived risk

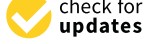



## 1. Introduction

Coping with sea-level rise as a result of climate change is one of the biggest societal challenges of this century as global mean sea-level rise resulting from thermal expansion (due to the warming of the oceans), loss of ice by glaciers, and ice sheets is accelerating. Studies have shown a higher possibility of sea level rise in the 21st century if the Antarctic and Antarctica ice sheets sections were to collapse [1–3]. Analysis of satellite altimetry data from 1993 to 2015 reveals that sea surface height increased nearly three times more than in previous years [4,5]. The fifth assessment report (AR5) of the Intergovernmental Panel on Climate Change predicted that the average global sea level will likely rise to between 28–68 cm and 52–98 cm by 2100 (RCP 2.6 and 8.5 respectively) based on process-based model projections [6,7]. Changes in sea-level rise have already impacted coastal communities by increasing the risk of flooding and/or erosion of coastal ecosystems and infrastructure [8–10]. For instance, in some low-lying coastal areas, especially in the eastern part of Ghana, the coastline is eroding by 20 m or more per year, resulting in loss of settlements and livelihood assets.

Sea-level rise adaptation strategies require regulations, plans, and measures that reduce risks and build resilience. The Intergovernmental Panel on Climate Change (IPCC),

in their fifth assessment report (AR5), posited a tripartite framework of retreat (moving away from the coast), protect (structural and soft measures), and accommodate (changes in human activities and infrastructure). Among the three basic categories of sea-level rise risk reduction approaches (protection, accommodation, and retreat) reviewed in the literature, managed retreat will be the only long-term adaptation approach in many flood-prone areas [11]. Managed retreat means rethinking coastal life and accepting that certain coastal infrastructures, neighborhoods, or even cities will have to be completely relocated [12]. This adaptation response can be carried out at different scales and with varying degrees of complexity. It may involve the relocation of a few vulnerable homes, a community, or a large city. It can be even more complicated when it involves relocating inhabitants of an island to a new country.

An important issue with relocation is the cooperation of the affected population, which is sometimes difficult. Relocation of low-lying areas is anticipated worldwide, especially in areas where sea-level rise causes flooding and coastal erosion, reduces arable land, depletes groundwater supplies, destroys infrastructure, and endangers human lives and well-being [13,14]. Some countries such as Fiji, Mozambique, and the Solomon Islands have developed and included planned relocation strategies as part of National Adaptation Programmes of Action (NAPAs) in their various countries [15,16]. Over the years, the Ghanaian government has used gabions and boulders to protect some major cities and towns from sea-level rise. However, in the most vulnerable coastal rural areas, relocation of entire communities is often proposed, but residents are usually unwilling to relocate due to various reasons reported by [17]. Relocation costs are also underestimated because social and cultural values are not always completely considered. Despite the government's prioritization of adaptation strategies, the understanding of individual-level adaptation, especially in coastal rural areas, has been inadequate. Climate change adaptation strategies will be ineffective unless they are implemented in the context of household perceptions of climate change risk and self-efficacy in hazard reduction [18]. Although several studies have highlighted the factors that influence climate change adaptation action globally [19–21], there are few studies on people's attitudes toward sea-level rise adaptation strategies [13]. More importantly, there are few studies linking behavioral aspects to adaptation to sea-level rise, and no such studies have been conducted in Ghana. Several studies have adopted the Protection Motivation Theory (PMT) as the foundation of a socio-cognitive model to describe individual adaptation behavior in response to climate change [22–25]. Protection Motivation Theory explains the cognitive process people go through when assessing their own ability to avoid a particular risk. Drawing on the PMT, this study examines relocation intention in three coastal rural communities in Ghana, each affected by coastal erosion and flooding. The study aims to understand the relocation intention of rural coastal residents in anticipation of sea-level rise. The study provides insight into the behavioral aspects of implementing managed retreat as an adaptation response to reduce the impacts of sea-level rise. Additionally, the study will highlight the application of PMT in the context of climate change adaptation.

## 2. Conceptual Framework

*Factors Influencing Household-Level Adaptation Action to Climate Change*

The study developed a theoretical model based on an extended version of Protection Motivation Theory that comprises a combination of cognitive, socio-demographic, and physical factors that may influence the intention of coastal rural residents to relocate (see Figure 1). PMT was originally formulated by [26] based on the work of [27]. It was first used in health threat and safety assessments and later used beyond health-related issues to a more general theory to solve problems like political issues, environmental issues, injury prevention, and other social issues. The PMT suggests that individuals protect themselves based on the perceived probability of the occurrence of an event, the perceived severity of a threatening event, the efficacy of the recommended preventive behavior, perceived self-efficacy, and finally, the response cost [28]. According to [26], people balance different

risks and potential benefits based on their motivation to protect themselves from threats such as natural disasters, nuclear explosions, and global climate change. As a result, PMT assumes that people's decisions to engage in risk-reducing behaviors are based on two cognitive processes: threat appraisal and coping appraisal.

Threat appraisal is a cognitive process that refers to the perceived expectation of being exposed to a particular threat/risk. The coping appraisal, on the other hand, comprises self-efficacy (an individual's perception of their capability to perform the behaviors) and response efficacy (the perceived effectiveness of the recommended risk-preventative behaviors). Threat appraisal and coping appraisal mediate the effects of the components of fear appeals on attitudes by arousing an individual's motivation to protect and can influence individuals to perform adaptation actions such as relocation in anticipation of sea-level rise. PMT has been used by many researchers in natural hazards, disasters, and pro-environmental behaviors. For instance, [25] conducted a study in Vietnam based on the conceptual framework of protection motivation proposed to investigate the determinants of household flood protective strategies and risk perception using data from a household-level survey. Another study was conducted on household adaptation and intention to adapt to coastal flooding using PMT in Greece in 2013. The author [29] explored the existing adaptation behavior of the coastal households, identified determinants that influence the precautionary behaviors, and assessed the intention of adaptation of the households in the future.

In addition to the traditional components, the extended version of the PMT used in this study includes three additional components that were identified through the literature review (see Figure 1). These include risk perception, compositional, and contextual factors. Compositional factors are linked to sociodemographic traits of an individual or a group [30]. These factors were subdivided into biosocial and socio-cultural factors. Biosocial factors are underlying biological or physical qualities of people that are fixed from birth and cannot be changed, whereas socio-cultural factors are related to beliefs, values, and way of life [31]. Contextual factors include biophysical attributes (slope, elevation, distance to hazard-prone areas.). An increasing body of literature examines individual responses to sea-level rise impacts such as flooding, storm surge, erosion, and other related risks, with the majority focusing on determining the relationship between these factors and adaptation efforts. For example, several studies have shown that risk perception positively influences individual adaptation behavior. A person with a high risk perception is more likely to undertake adaptation measures [32–35]. However, in their study on the factors that motivate rural households to adapt to climate change, [36] established that adaptation appraisal rather than risk perception is a better predictor of climate change adaptation. In comparison to risk perception, the relationship between socio-demographic variables and protective behavior adoption is significantly less clear. Notwithstanding, several studies have identified various socio-demographic factors related to climate change adaptation efforts such as educational level [37], income [33], age [35,37], gender [38], and location in terms of rural or urban setting [39]. Hazard experience is also considered to have a significant influence on risk recognition and appears to be a significant component in individual adaptation behavior [40]. For instance, individual views of flooding resilience were explored in four communities in Birmingham and London, UK, by [41], who found that people's social responsibility for adaptation measures was influenced by their experience with floods as well as other demographic factors. Other studies also confirmed that hazard experience was positively associated with adaptation efforts in their respective studies [42,43]. A similar study by [44] showed that experiencing flood hazards did not motivate citizens to take more proactive adaptation measures. The capacity of human systems to adapt to a changing climate is linked to characteristics of the physical environment. Physical factors such as a lack of high elevation to relocate to, for example, can limit relocation [45]. Also, proximity to hazards can also limit adaptation efforts. Studies have revealed that hazard proximity can influence risk perception, and people living close to hazard areas will adopt coping strategies to reduce the risk [46,47]. In addition, studies have been conducted to

assess the relationship between proximity to hazards and adaptation efforts. However, their findings have been inconsistent; although some researchers have discovered a positive link, others have not. In their study about analyzing risk perception and precautionary behavior, [22] discovered that the distance to a river or waterbody had only a minor impact on people's current risk reduction efforts, whereas [48] reported contradictory findings.

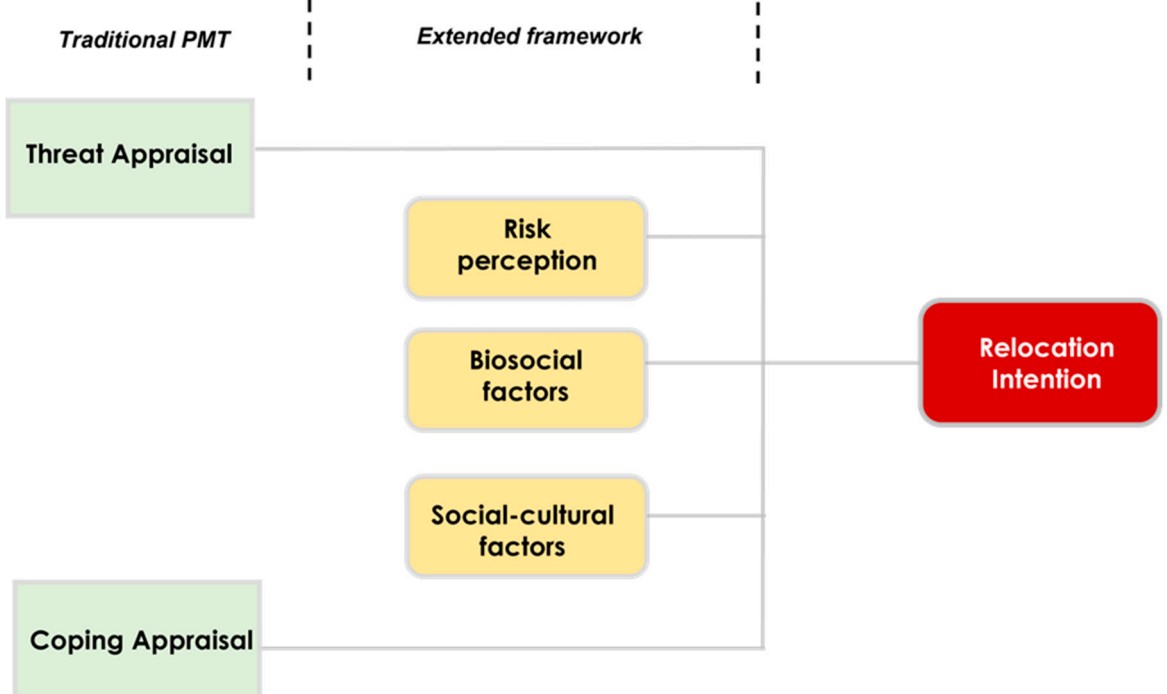

**Figure 1.** An extended framework of Protection Motivation Theory. Source: Modified from [33].

## 3. Materials and Methods

### 3.1. Study Area

The study was carried out in three rural coastal communities in the Greater Accra and Western Regions of Ghana (Figure 2). The three selected rural communities, namely Sanwoma (4°54′08.9″ N; 2°16′14.1″ E), Anlo Beach (5°14′5.4″ N; 1°36′28.9″ W) and Glefe-Wiaboman (5°31′3.2″ N; 0°17′21.2″ W) were identified as coastal erosion and flooding hotspots with reports of loss of settlements and livelihood assets [40–42]. The proximity of these three communities to some major estuaries and wetlands makes them highly susceptible to impacts from sea-level rise. These communities were selected based on several criteria: located in a rural area, estuarine communities, and documented reports of frequent coastal flooding and erosion. The study communities are low-lying coastal plains located at the mouth of the Ankobra (Sanwoma), the Pra (Anlo Beach), and the Densu River (Glefe-Wiaboman), with portions of the land occasionally flooded and often hit by tidal waves leading to the displacement of residents. The climate of the study communities falls within the dry equatorial climate with double rainfall maxima. The major rainy season is from March to June and the minor rainy season begins in September and ends in November, followed by a dry season from December to March. The temperatures in Ghana's coastal savannah region are high throughout the year, with an annual mean temperature of 26.5 °C. The average monthly temperature is between 24.5 °C (August) and 28 °C (March) and the average daytime temperature is 30 °C (August). Humidity is generally high (65–95%) but is lower during the warmer months, especially in January with dry northeast Harmattan winds. The vegetative cover is made up of coastal strands and mangroves and freshwater vegetation. Mangroves of the genera *Avicennia*, *Rhizophora*, and *Laguncularia* fringe the banks of the estuaries. The adjoining marshland mostly has the saltwater grass *Paspalum vaginatum* (Poaceae) as the main vegetation. The mangrove trees are highly exploited as the

main source of firewood for cooking and smoking fish in the communities, thereby resulting in the degradation of the mangrove forest. Fishing (beach seining) and fish mongering are the major livelihood activities in the study communities, running from mid-July to late April. Fishing is done mostly by men while women are involved in fish processing. Subsistence farming becomes the predominant occupation after April, lasting for about three months in the off-fishing season. The community switches back to fishing in mid-July or early August for the main fishing season. In some of the communities, however, some residents are involved in agricultural activities and grow crops such as cowpea, sweet potatoes, maize, okro (okra), tomatoes, pepper, etc. Additionally, most of the inhabitants in the study communities are engaged in some livestock rearing; animals such as cattle, sheep, goats, pigs, and poultry can be found in the community.

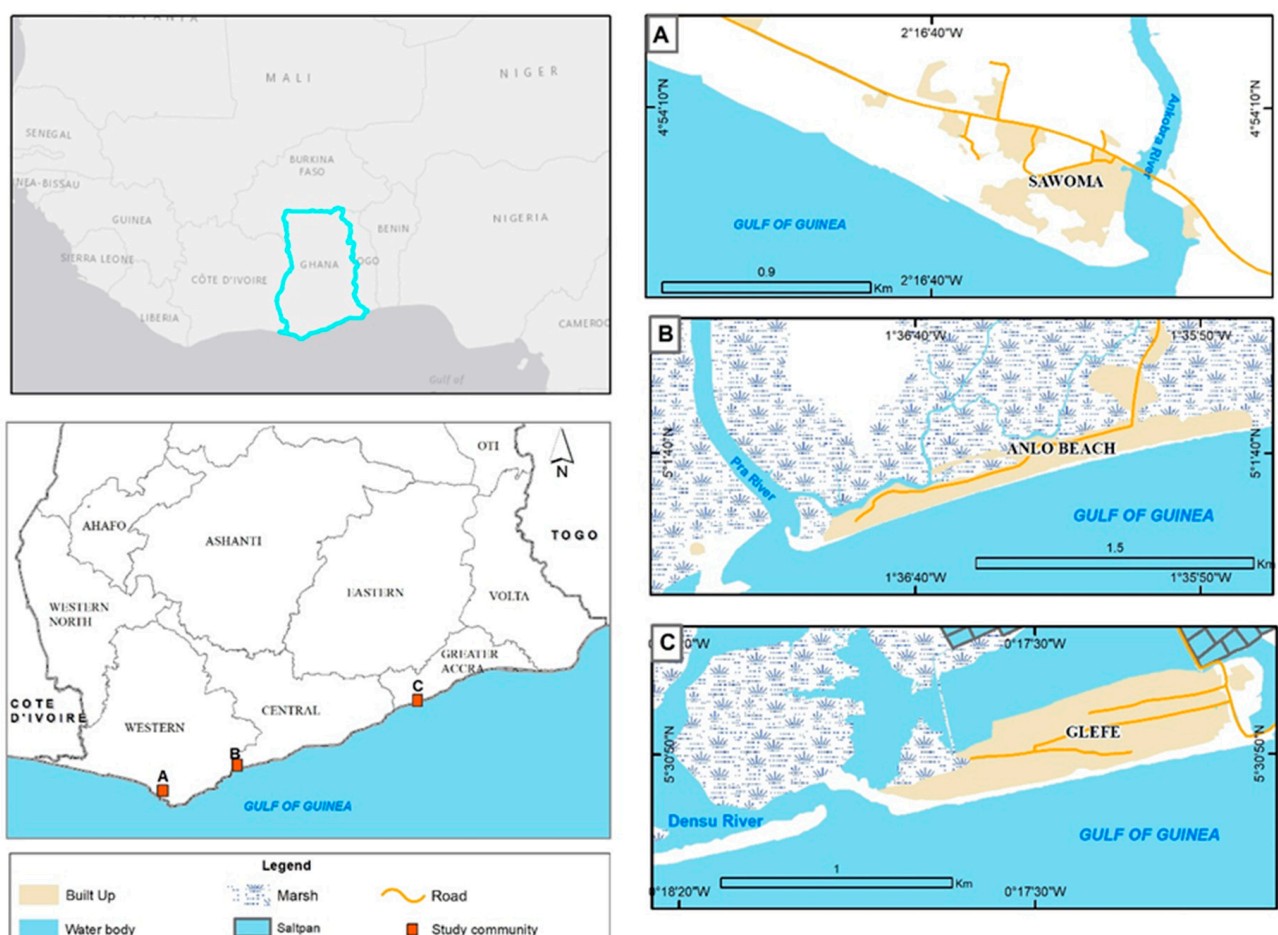

**Figure 2.** Map showing the location of study communities. Source: Authors.

### 3.2. Questionnaire Design and Data Collection

The study was based on primary data from a structured questionnaire. An initial draft of the instrument was designed and pretested to see the practicalities in administering the survey and identify possible challenges that could be faced. The outcome variables considered in this study was relocation intention in anticipation of sea level rise, and it was represented as binary variable, with '1' representing 'will relocate' and '0' representing 'will not relocate'. The predicted/independent variables considered in the study included variables stipulated in the traditional components and the extended version of the PMT. These include cognitive, contextual factors and compositional (biosocial and socio-cultural) factors (Table 1). The structured questionnaire was administered using the KoBoTool box mobile application. The instrument was mainly designed based on the work of [35] and also information from the three Focus Group Discussion (FDG) conducted in the study

communities. The survey targeted heads of households in the various communities. A total of 359 respondents were randomly selected from a target population of 1468 household heads based on the sampling technique of [49]. The multistage sampling method was employed to select the respondents for the study. For the first stage, a cluster sampling procedure was applied where the study communities were divided into clusters using a georeferenced hexagon with a side of a 2000 m$^2$ grid (Figure 3). The number of buildings (digitized on-screen from the 2021 UAV image) was chosen as the unit for the proportional allocation of respondents for the hexagon grid. The number of respondents in a grid was calculated based on the number of buildings, and a grid with only one building was not included in the sample. A simple random sampling was used to select the specific building in the grid to be visited at the third stage. In order to know the particular buildings selected and to be visited, the spatial extent of all of the selected buildings was converted to shapefile and loaded onto the SW Map App on a mobile phone, which led the researcher to the actual building in the field. Lastly, a convenience sampling technique was used to select the household head to be interviewed in the selected building. However, buildings with no occupants at the time of the interview and those with no adult in charge of the household in the absence of the household head were excluded. As a result, the next building with a household head was chosen as a replacement.

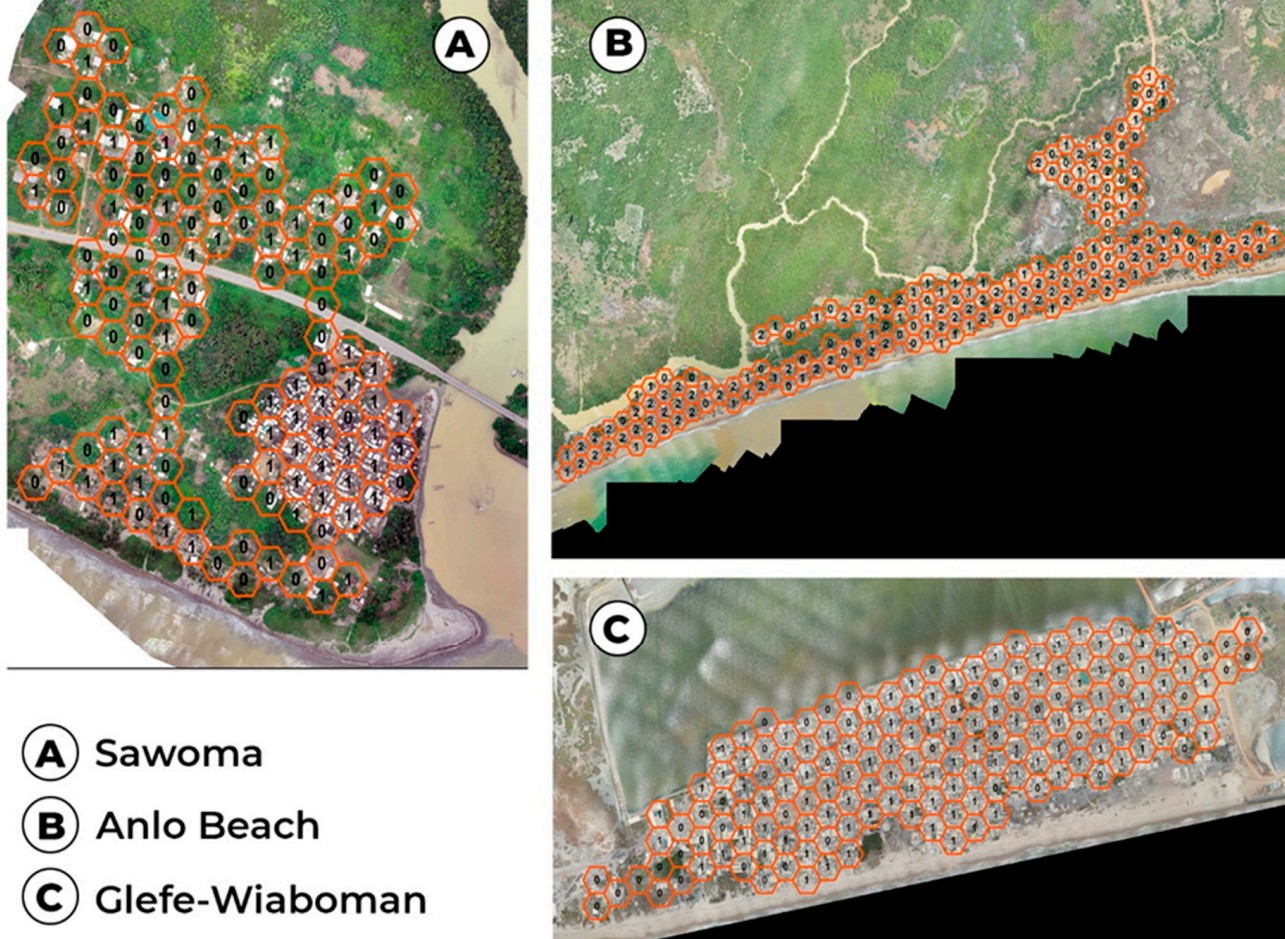

**Figure 3.** Sample grids for the three study communities.

**Table 1.** Measured factors and their coding scheme.

| Factors | Item | Measurements | Coding Scheme |
|---|---|---|---|
| Relocation Intention | RI | Relocation decision | Binary scale (1- Will relocate, 0- Will not relocate) |
| Risk perception | RP1 | Sea-level rise is taking place | Likert scale: 1–5 (strongly agree to strongly disagree) |
| | RP2 | Sea level rise poses a danger to the natural environment | |
| | RP3 | Sea-level rise poses a danger to the built environment | |
| Threat appraisal | TA1 | Sea-level rise impacts become too frequent and destructive | 5-point scale: 1 (not important at all) to 5 (very important) |
| | TA2 | Safety of myself and/or my family | |
| | TA3 | Neighborhood, friends, and/or family decide to leave the area | |
| | TA4 | Property is severely damaged | |
| | TA5 | No provision of adaptation measures | |
| Coping Appraisal | CA1 | Relocation cost | 5-point scale: 1 (not important at all) to 5 (very important) |
| | CA2 | Distance to current workplace | |
| | CA3 | Job opportunities at the new location | |
| | CA4 | Social and family ties | |
| Biosocial factors | BF1 | Sex of respondent | 1 = Female, 2 = Male |
| | BF2 | Age of respondent | 1 = <35, 2 = 35–55, 3 = >55 |
| | BF3 | Experienced damage as a result of sea-level rise | Binary scale (1–yes, 0–no) |
| Socio-cultural factors | SF1 | Educational level of respondent | 1 = No formal education, 2 = Primary, 3 = JHS/Middle 4 = SHS/Voc/Tech and above |
| | SF2 | Income level of respondent | 1 = >GHC 100, 2 = GHC 101-500, 3 = GHC 501-999, 4 = GHC 100 and above |
| Contextual factors | | Elevation | 1 = <4 m 2 = 4.01–7 m, 3 = 7.01 m and above |
| | | Location of house from shoreline | 1 < 100 m, 2 = 101–400 m, 3 = 401–700 m, 4 = Above 700 m |

JHS—Junior High School; SHS—Senior High School; Voc—Vocational School; Tech—Technical School; GHC—Ghana Cedis.

### 3.3. Data Processing and Analysis

The statistical analyses were carried out with IBM SPSS version 24. Descriptive statistics were used to quantitatively describe and summarize the characteristics of the explanatory variables. Secondly, the cognitive factors (risk perception, threat appraisal, and coping appraisal) were subjected to Exploratory Factor Analysis (EFA) to examine the strength and relationship between measured variables before including them in the model. The Kaiser–Meyer–Olkin (KMO) value was 0.782, which was higher than the crucial limit of 0.7 [50]. The Bartlett sphericity test yielded a significant value of $p = 0.000$, indicating that the original dataset was suitable for factor analysis. In the EFA process, constructs were extracted from all original items using principal component analysis with varimax rotation and factor loadings greater than 0.6. Three items (TA3, TA5, and CA1), which had a factor loading lower than 0.6, were deleted. Cronbach's values for internal validity were determined to test the revised scale's reliability. Cronbach's values of all derived variables were around or greater than 0.7, ranging from 0.801 to 0.901 (See Table 2). Cronbach's value should be higher than 0.700, according to [50]. Thus, it can be inferred that all cognitive variables in the modified scale were internally consistent and reliable enough to be included in the model. The relationship between the dependent variable (relocation intention) and the explanatory variables was examined using both univariate and multivariate statistical techniques. The association between the dependent variable (relocation intention) and compositional and contextual factors was examined using Pearson's chi-squared test. Cramer's V statistic was employed to determine how strongly the variables were associated. Analysis of variance (ANOVA) was also carried out to assess the relationship between cognitive factors and the dependent variable. Post hoc tests were employed to compare the differences between the cognitive factors, compositional, and contextual factors. With regards to the sex of the respondent and hazard experience variable, the mean difference comparison was done using an independent *t* test. The factors that influence relocation intention were modeled using ordered logistic regression. Binary logistic regression was used for the analysis since the dependent variable (relocation intention) is dichotomous, with '1' representing 'will relocate' and '0' representing 'will not relocate'. Thus, the regression model is expressed in terms of the logit instead of Y as shown in equation 1. Subsequently, four models were run: model 1 examined the influence of the cognitive factors, model 2 looked at cognitive and biosocial factors, model 3 focused cognitive, biosocial, and sociocultural factors, and model 4 combined cognitive, biosocial, sociocultural, and contextual factors.

$$\text{logit} = L_i = B_0 + B_{1 \times 1} + \ldots + B_K X_K \tag{1}$$

**Table 2.** Varimax-rotated component analysis factor matrix and Cronbach's $\alpha$ values for the cognitive'variables.

| Constructs | Items | Main factors | | | Cronbach Alpha |
|---|---|---|---|---|---|
| | | **1** | **2** | **3** | |
| Risk perception | RP1 | 0.907 | | | 0.901 |
| | RP2 | 0.906 | | | |
| | RP3 | 0.915 | | | |
| Threat appraisal | TA1 | | 0.893 | | 0.861 |
| | TA2 | | 0.922 | | |
| | TA4 | | 0.847 | | |
| Coping appraisal | CA2 | | | 0.835 | 0.801 |
| | CA3 | | | 0.817 | |
| | CA4 | | | 0.765 | |

Kaiser-Meyer-Olkin Measure of Sampling Adequacy = 0.782 and Bartlett's Test of Sphericity $p = 0.000$.

## 4. Results

### 4.1. Descriptive Statistics

Table 3 summarizes the respondents' socio-demographic characteristics. As the communities studied are not homogeneous, it is important to understand their socio-economic composition in order to assess their behavior towards climate adaptation measures. Other variables considered in the study are also summarized using figures.

**Table 3.** Socio-demographic characteristics of respondents.

| Background Characteristics | Community | | | | | | | |
|---|---|---|---|---|---|---|---|---|
| | Sanwoma | | Anlo Beach | | Glefe-Wiaboman | | Total | |
| | N | % | N | % | N | % | N | % |
| Community | 64 | 17.8 | 193 | 53.8 | 102 | 28.4 | 359 | 100 |
| Sex | | | | | | | | |
| Male | 23 | 13.9 | 96 | 58.2 | 48 | 27.9 | 165 | 46.0 |
| Female | 41 | 21.1 | 97 | 50.0 | 58 | 28.9 | 194 | 54.0 |
| Age (years) | | | | | | | | |
| <35 | 22 | 17.2 | 55 | 43.0 | 51 | 39.8 | 128 | 35.7 |
| 35–55 | 33 | 18.8 | 98 | 55.7 | 45 | 25.6 | 176 | 49.0 |
| >55 | 9 | 16.4 | 40 | 72.7 | 6 | 10.9 | 55 | 15.3 |
| Educational level | | | | | | | | |
| No formal education | 13 | 20.0 | 46 | 70.8 | 6 | 9.2 | 65 | 18.1 |
| Primary | 19 | 19.4 | 67 | 68.4 | 12 | 12.2 | 98 | 27.3 |
| JHS/Middle | 25 | 16.2 | 66 | 42.9 | 63 | 40.9 | 154 | 42.9 |
| SHS/Voc/Tech | 7 | 16.7 | 14 | 33.3 | 21 | 50.0 | 42 | 11.7 |
| Average monthly income | | | | | | | | |
| >GHC 100 | 8 | 22.2 | 17 | 47.2 | 11 | 30.6 | 36 | 10 |
| GHC101–500 | 44 | 23.0 | 104 | 54.5 | 43 | 22.5 | 191 | 53.2 |
| GHC 501–999 | 8 | 10.3 | 48 | 61.5 | 22 | 28.2 | 78 | 21.7 |
| <GHC 1000 | 4 | 7.4 ss | 24 | 44.4 | 26 | 48.1 | 54 | 15.0 |

#### 4.1.1. Socio-Demographic Characteristics of Respondents

Out of the 359 respondents from the three coastal rural communities, 64 (17.8%) were from Sanwoma, 193 (53.8%) from Anlo Beach, and 102 (28.4%) from Glefe-Wiaboman (Table 3). The percentages of male and female respondents were nearly equal among the 359 respondents, with 46.0% and 54%, respectively. Majority of respondents (49.0%) were between the ages of 35 and 55. In terms of educational level, the majority of the respondents had completed middle school/junior high school (42.9%). Most of respondents (53.2%) earned GHC101–500 every month. Only 10% of the respondents earned less than GHC 100 per month.

#### 4.1.2. Risk Perception, Threat, and Coping Appraisal

As shown in Figure 4, communities had some differences in terms of cognitive factors. Anlo Beach and Sanwoma had a score greater than 4.0 for all factors of threat appraisal except for TA3 (neighborhood, friends, and/or family decide to leave the area), for which Sanwoma scored less than 4.0. Glefe-Wiaboman, on the other hand, had a score of less than 4.0 for all threat appraisal factors (Figure 4a). For coping appraisal factors, Anlo Beach had a score greater than 4.0 for all the factors. Sanwoma had a score of less than 4.0 except for CA1 (relocation cost), with a score of 4 (Figure 4b). Anlo Beach and Sanwoma scored the highest mean score greater than 3 in all risk perception factors, whereas Glefe had a mean score less than 3 (Figure 4c).

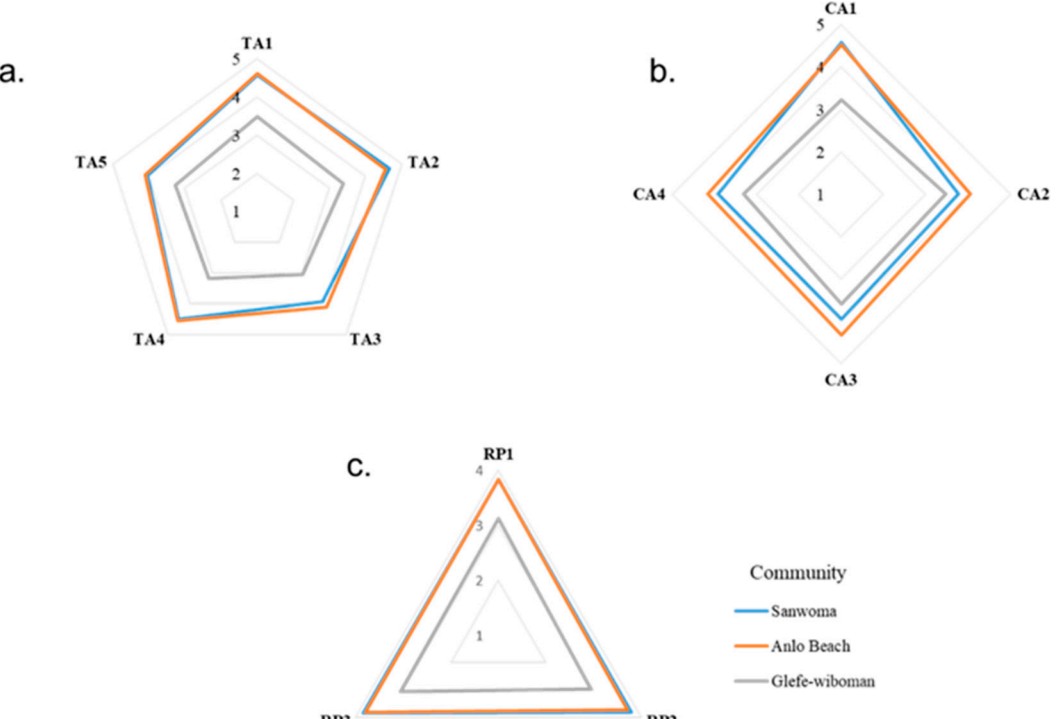

**Figure 4.** Scored values of factors associated with (**a**) threat appraisal (TA), (**b**) coping appraisal (CA), and (**c**) risk perception (RP).

Refer to Table 1 for details on (TA1–TA5; CA1–CA4, RP1–RP3).

### 4.1.3. Hazard Experience

Out of 359 respondents interviewed, 293 indicated that they have experienced hazards resulting from sea-level rise (Figure 5). Of these, 185 (63.1%) were reported in Anlo Beach, while 57 (19.5%) and 51 (17.4%) were in Sanwoma and Glefe-Wiaboman, respectively.

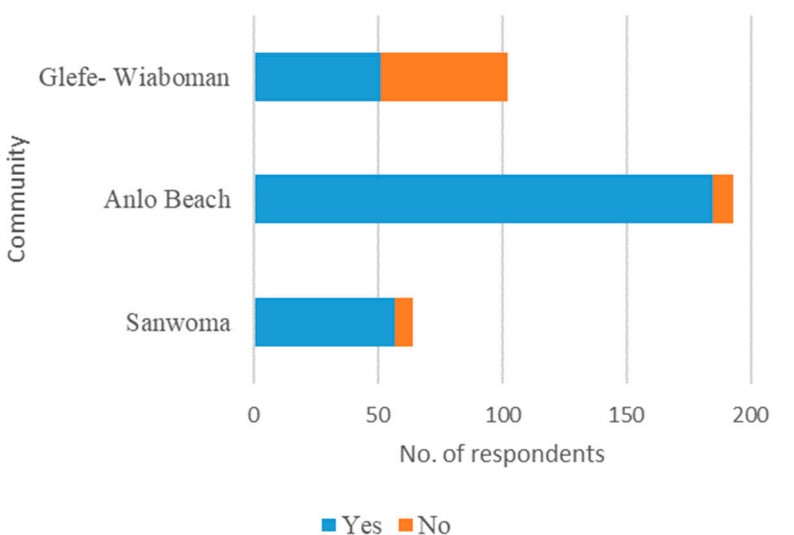

**Figure 5.** Respondents' experience of sea-level impacts (coastal erosion/flooding) in the study communities.

### 4.1.4. Proximity to Shoreline

Most of the respondents in study communities were affected by the flood and/or erosion depending on their proximity to the coastline. The results presented in Figure 6

indicated that 40% of the respondents in Sanwoma were located 101–400 m to the shoreline and 5% were located 700 m and beyond away from the shoreline. In the Anlo beach community, 88% of the respondents were less than 100 m away from the erosion risk areas, and 5% were within 401–700 m of the shoreline. A vast difference was noted in Glefe-Wiaboman, where 51% of the respondents were located within less than 100 m to the shoreline and 49% lived between 101 and 400 m of the shoreline.

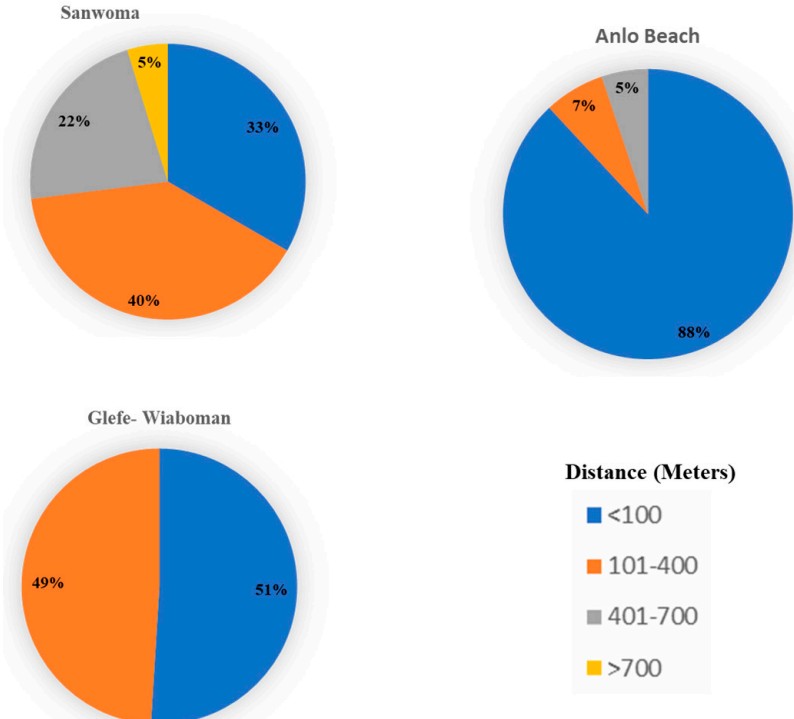

**Figure 6.** Respondents' proximity to shoreline in the study communities.

### 4.2. Measures of Association

Post hoc tests were conducted to find differences between the compositional/contextual factors and the cognitive factors (Table S1). The results indicated that the age group middle-aged adult and older adults were statistically associated with risk perception ($p < 0.008$ and $p < 0.005$ respectively) compared to younger adults. In terms of level of education, primary education had a statistically significant influence on coping appraisal ($p < 0.027$) compared to respondents with no formal education. Respondents within the income categories GHC 101–500 and GHC 501–999 had statistically significant influence on both threat appraisal and coping appraisal. Hazard experience of the respondents had statistically significant influence on risk perception ($p < 0.000$) and threat appraisal ($p < 0.019$). Respondents living at a distance between 100 and 300 m from shoreline showed a statistically significant association with risk perception ($p < 0.000$), threat appraisal ($p < 0.009$), and coping appraisal ($p < 0.015$). Distance of the respondents to flood risk areas had a statistically significant influence on both threat appraisal ($p < 0.002$) and coping appraisal ($p < 0.036$).

ANOVA was conducted to examine the association between the cognitive factors and relocation intention, whilst Pearson chi-squared and Cramer's V statistics were employed to assess the relationship between compositional/contextual factors and relocation intention. The results of the one-way ANOVA (Table 4) show that among the three cognitive factors, risk perception had a statistically significant relationship with relocation intention. Additionally, Pearson chi-squared and Cramer's V statistics (Table 5) indicated no association between relocation intention and the compositional and contextual factors.

**Table 4.** Analysis of variance (ANOVA) results of cognitive factors and relocation Intention.

| Variable | *p*-Value |
|---|---|
| Risk perception | 0.000 |
| Threat appraisal | 0.084 |
| Coping appraisal | 0.040 |

**Table 5.** Distribution of compositional and contextual variables by predictor variables.

| Variable | Relocation Intention | | |
|---|---|---|---|
| | **Will Relocate** | **Will Not Relocate** | **Inferential Statistics** |
| Sex of respondent | | | |
| Male | 147 | 18 | $\chi^2 = 0.79$, *p*-value = 0.456; Cramér's V = 0.15 |
| Female | 171 | 23 | |
| Age of respondent | | | |
| Young adult | 117 | 11 | |
| Middle-aged adult | 152 | 24 | $\chi^2 = 1.879$, *p*-value = 0.391; Cramér's V = 0.72 |
| Older adult | 49 | 6 | |
| Educational level | | | |
| No formal education | 55 | 10 | |
| Primary | 85 | 13 | $\chi^2 = 2.700$, *p*-value = 0.440; Cramér's V = 0.087 |
| JHS/Middle | 141 | 13 | |
| SHS/Voc/Tech and above | 37 | 5 | |
| Average monthly income | | | |
| >GHC 100 | 33 | 3 | |
| GHC 101–500 | 170 | 21 | $\chi^2 = 2.700$, *p*-value = 0.440; Cramér's V = 0.087 |
| GHC 501–999 | 71 | 7 | |
| <GHC 1000 | 44 | 10 | |
| Elevation | | | |
| >4 m | 233 | 30 | |
| 4–9 m | 73 | 11 | $\chi^2 = 1.780$, *p*-value = 0.411; Cramér's V = 0.070 |
| <9 m | 12 | 0 | |
| Distance to shoreline | | | |
| >100 m | 134 | 12 | |
| 100–400 m | 159 | 27 | $\chi^2 = 3.671$, *p*-value = 0.160; Cramér's V = 0.101 |
| <400 | 25 | 2 | |
| Hazard experience | | | |
| Yes | 264 | 29 | $\chi^2 = 3.654$, *p*-value = 0.440; Cramér's V = 0.101 |
| No | 54 | 12 | |

$\chi^2$ = Pearson chi-squared.

### 4.3. Factors Affecting Relocation Intention

The relationship between relocation intention and the key predictors (cognitive factors) compositional and contextual factors, were examined using four different models in the multivariate analysis. The models employed were cognitive factors (model 1), biosocial factors (model 2), sociocultural factors (model 3), and contextual factors (model 4). Table 5 presents the proportional odds ratios, robust standard errors, probability values, and confidence intervals for the cognitive factors, compositional, and contextual components.

The results of model 1 indicate that risk perception (*p*-value < 0.005) and threat appraisal (*p*-value < 0.05) among the cognitive factors have a positive significant relationship with relocation intention. This suggests that households who believe that sea-level rise is taking place and poses danger to both natural and built environments are more likely to consider relocating. In model 2, where we controlled for biosocial factors, found that risk perception and threat appraisal continued to have a positive significant relationship with relocation intention. Furthermore, households headed by middle-aged adults were found

to be 56% (*p*-value < 0.05) less likely to relocate compared to households with young adult heads. However, there was no significant relationship between sex and relocation intention. The results of model 3 (sociocultural) showed no substantial differences from model 2 concerning risk perception, threat appraisal, sex, and age, except for minor variations in the proportional odds ratios. Moreover, the sociocultural factors, including education and household income, had no significant relationship with relocation intention. In the final model, where contextual factors including hazard experience and distance of house from shoreline were controlled for, there were slight changes in the proportional odds ratios for the variables, including risk perception, threat appraisal, and age, that had a significant relationship with relocation intention in model 3. Moreover, the study found that household monthly income, which was not significant in model 3, became a significant predictor in the contextual model. Households with a monthly income of 1000 cedis and above were 70% less likely to relocate (*p*-value < 0.05) compared to those with a monthly income below 100 cedis. However, neither of the two contextual factors exhibited any significant association with relocation intention.

## 5. Discussion

The study reveals some new trends in how coastal rural communities react to long-term threats arising from the impacts of climate change. This provides insights into the behavioral aspect of implementing managed retreat as an adaptation strategy to curb the impacts of sea-level rise. According to Protection Motivation Theory and other previous studies [29,35], adaptation behavior is linked to cognitive variables such as risk perception, threat, and coping appraisal. In this study, risk perception appears to be a significant factor for explaining relocation intention. A prominent role of perceived sea level rise risk in promoting adaptation was found by [29,35]. Since risk perception increases the intensity of adaptation, it is important to emphasize this to encourage coastal rural households to take protective measures, and one way to improve risk assessment would be to educate them on the impending sea-level rise impacts. Contrary to the study by [36], risk perception as shown in this study is a better predictor of climate change adaptation compared to adaptation appraisal. In addition, the study also established that threat appraisal is a better predictor for relocation intention than coping appraisal. This echoes the findings of [35]. Indeed, in Table 6, model 1 shows that the perceived risk and the perceived expectation of being exposed to the risk in the study communities positively influence respondents' intentions to relocate, but capacity to perform risk-preventative behaviors does not significantly influence these intentions. The study also confirms the conclusion drawn by researchers such as [35,48] that the influence of biosocial factors on climate change adaptation action is mixed and varies between contexts. In this study, age appears not to be a significant factor for explaining adaptation behavior. On the other hand, age was found to have a strong positive association with risk perception (Table S1). In general, the older the respondent, the higher the sea-level rise risk perception level they have. This may be because older respondents have experienced many historical sea-level rise impacts and they are accountable for the safety of their family. Researchers such as [35] further argued that in the event of a sea-level rise disaster, young people are more likely to stay since they have stable income sources and strong social ties. As a result, letting go of these areas of one's life and relocating to a new location might be difficult.

**Table 6.** Ordered logistic regression model showing the relation between relocation intention and household characteristics.

| Variables | Odds Ratio | Robust SE | *p*-Value | Conf. Interval | |
|---|---|---|---|---|---|
| Model 1: Cognitive Factors | | | | | |
| Risk perception | 1.495 | 0.179 | **0.001** | 1.182 | 1.890 |
| Threat Appraisal | 1.334 | 0.160 | **0.017** | 1.054 | 1.688 |
| Coping Appraisal | 1.304 | 0.190 | 0.068 | 0.980 | 1.734 |

**Table 6.** *Cont.*

| Variables | Odds Ratio | Robust SE | *p*-Value | Conf. Interval | |
|---|---|---|---|---|---|
| Model 2: Model 1 + Biosocial factors | | | | | |
| Risk perception | 1.572 | 0.200 | **0.000** | 1.225 | 2.018 |
| Threat Appraisal | 1.327 | 0.165 | 0.023 | 1.040 | 1.692 |
| Coping Appraisal | 1.290 | 0.182 | 0.071 | 0.979 | 1.700 |
| Sex (ref: Female) | | | | | |
| Male | 1.202 | 0.350 | 0.527 | 0.679 | 2.128 |
| Age (ref: Young adult) | | | | | |
| Middle-aged adult | 0.440 | 0.155 | **0.020** | 0.221 | 0.876 |
| Older adult | 0.919 | 0.490 | 0.875 | 0.323 | 2.614 |
| Model 3: Model 2 + Socio-cultural factors | | | | | |
| Risk perception | 1.633 | 0.215 | **0.000** | 1.261 | 2.115 |
| Threat Appraisal | 1.359 | 0.177 | **0.019** | 1.052 | 1.754 |
| Coping Appraisal | 1.208 | 0.178 | 0.198 | 0.906 | 1.611 |
| Sex (ref: Female) | | | | | |
| Male | 1.141 | 0.346 | 0.664 | 0.629 | 2.068 |
| Age (ref: Young adult) | | | | | |
| Middle-aged adult | 0.397 | 0.154 | **0.017** | 0.185 | 0.850 |
| Older adult | 0.725 | 0.402 | 0.562 | 0.245 | 2.148 |
| Education (ref: No formal education) | | | | | |
| Primary | 1.370 | 0.608 | 0.478 | 0.574 | 3.270 |
| Middle School/JHS | 1.383 | 0.615 | 0.467 | 0.578 | 3.308 |
| Secondary School and above | 1.290 | 0.717 | 0.646 | 0.434 | 3.832 |
| Household monthly income (GHC) (ref: below 100) | | | | | |
| 101–500 | 0.394 | 0.218 | 0.093 | 0.133 | 1.167 |
| 501–999 | 1.014 | 0.692 | 0.984 | 0.266 | 3.862 |
| 1000 and above | 0.303 | 0.189 | 0.056 | 0.089 | 1.030 |
| Model 4: Model 3+ Contextual factors | | | | | |
| Risk perception | 1.421 | 0.223 | **0.025** | 1.045 | 1.933 |
| Threat Appraisal | 1.316 | 0.175 | **0.039** | 1.014 | 1.707 |
| Coping Appraisal | 1.178 | 0.171 | 0.260 | 0.886 | 1.565 |
| Sex (ref: Female) | | | | | |
| Male | 1.141 | 0.352 | 0.668 | 0.623 | 2.090 |
| Age (ref: Young adult) | | | | | |
| Middle-aged adult | 0.403 | 0.155 | **0.018** | 0.190 | 0.857 |
| Older adult | 0.693 | 0.380 | 0.504 | 0.237 | 2.030 |
| Education (ref: No formal education) | | | | | |
| Primary | 1.473 | 0.676 | 0.398 | 0.599 | 3.620 |
| Middle School/JHS | 1.547 | 0.702 | 0.336 | 0.636 | 3.765 |
| Secondary School and above | 1.519 | 0.908 | 0.485 | 0.471 | 4.900 |
| Household monthly income (GHC) (ref: below 100) | | | | | |
| 101–500 | 0.375 | 0.200 | 0.067 | 0.132 | 1.069 |
| 501–999 | 0.995 | 0.661 | 0.994 | 0.271 | 3.657 |
| 1000 and above | 0.302 | 0.184 | **0.049** | 0.092 | 0.995 |
| Hazard Experience (ref: No) | | | | | |
| Yes | 1.704 | 0.683 | 0.184 | 0.777 | 3.739 |
| Elevation | 1.010 | 0.100 | 0.918 | 0.832 | 1.226 |
| Distance of house from Shoreline (ref: below 100 m) | | | | | |
| 100–300 m | 0.655 | 0.238 | 0.244 | 0.321 | 1.335 |
| Above 300 m | 0.862 | 0.469 | 0.785 | 0.297 | 2.502 |

People with higher education should be more likely to pursue individual-level adaptation strategies in theory but the results in Table 6 indicated no association between relocation intention and education in the three rural coastal communities. However, studies have also reported a strong association between education and risk reduction behavior [29] and climate change action [51]. According to these studies, higher-educated persons were

less likely to adapt to climate change because they were more likely to understand issues of climate change and they also believe that it is the government's obligation to undertake high-cost adaptation strategies while they are able to implement low-cost and low-effort preventative steps. In terms of the respondents' monthly income, there was a significant relationship between income and relocation intention. This study revealed that high-income households were more likely to relocate compared to lower-income households, as they can afford the cost of relocation and also take other adaptation measures since they have more assets to protect themselves from sea-level rise disasters. Similar conclusions were also drawn by [51]. As seen in Figures 5 and 6, the majority of the respondents have experienced hazards in their lifetime and also live close to the shoreline. This was not surprising as these communities were situated along major estuaries and wetlands, making them highly susceptible to impacts from sea-level rise. In Anlo Beach, for example, the community is flooded for several weeks by seawater twice every year, destroying properties and obstructing economic activities. In July 2009 alone, 78 houses were destroyed, rendering several inhabitants homeless [52]. Despite these events, these hazard experiences and proximity to risk areas do not seem to influence their intention to relocate to a new area. However, these factors significantly influence the cognitive factors, as indicated in Table S1. Although the study provided important insights into the factors influencing the relocation of coastal rural households, a few limitations should be noted. First, this study only looked at residents' relocation intention, which is the most aggressive adaptation option to curb sea-level rise impacts. Individual perception of sea defense, beach nourishment, and other sea-level rise adaptation strategies are equally significant. A comparison of different strategies' perspectives is fascinating and can add greatly to the climate change resilience literature. Secondly, the factors considered in the extended version of the PMT may be limited. Finally, since some factors considered played a mediating role between the cognitive variables and relocation intention, the study could be extended further in follow-up research.

## 6. Conclusions

Using an extended version of Protection Motivation Theory, this study highlighted a range of factors that influence relocation intention of rural coastal residents in anticipation of sea-level rise. The results of this study showed that apart from the cognitive factors, compositional factors such as household age and income were more important for predicting the relocation intention of coastal rural communities in Ghana. Contextual factors such as hazard experience and proximity to shoreline did not appear to be significant in influencing residents' relocation intention, which was explained by the fact that most of the households were already used to sea-level rise impacts such as erosion and flooding. As a result, strategies to relocate these rural communities should focus on cognitive as well as compositional characteristics and, in particular, promote household-level adaptation as a viable and cost-effective approach to responding to sea-level rise impacts. This implies that disseminating information about the various aspects of impact of sea level rise can be utilized to create strategies and programs that will encourage relocation action. Thus, increased information dissemination by the government and civil society organizations could motivate households to relocate. The campaigns should emphasize the effects of future sea-level rise impacts on communities, increasing household self-confidence in adaption strategies and educating people about the benefits of relocation at the community level. Additionally, structural and non-structural measures are required for improving socio-economic capacity of coastal rural communities in order to adopt this adaptation strategy. This can be achieved through the provision of alternative livelihoods as well as making relocation cost-effective and affordable.

**Supplementary Materials:** The following supporting information can be downloaded at: https://www.mdpi.com/article/10.3390/cli11050110/s1, Table S1: Multiple Comparisons between compositional/contextual factors and cognitive factors.

**Author Contributions:** Conceptualization, R.A., D.J. and N.A.B.K.; methodology, R.A., D.J. and N.A.B.K.; software, R.A.; validation, D.J., N.A.B.K. and A.A.O.; formal analysis, R.A.; data curation, R.A.; writing—original draft preparation, R.A.; writing—review and editing, D.J., N.A.B.K. and A.A.O.; supervision, D.J. and N.A.B.K. All authors have read and agreed to the published version of the manuscript.

**Funding:** This study is part of PhD research funded by the German Federal Ministry of Education and Research (BMBF) through the West African Science Service Center on Climate Change and Adapted Land Use (WASCAL). The APC was funded by the Africa Centre of Excellence in Coastal Resilience (ACECoR), University of Cape Coast (Grant Number 6389-G).

**Data Availability Statement:** The data presented in this work are available upon request from the corresponding author.

**Acknowledgments:** Special appreciation is expressed to Bernard Ekumah for his support.

**Conflicts of Interest:** The authors declare no conflict of interest.

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
