# Peer review of "Adapting to Changing Climate: Understanding Coastal Rural Residents’ Relocation Intention in Response to Sea Level Rise"

_climate, doi:10.3390/cli11050110_

Round 1

Reviewer 1 Report

In my opinion, the work is interesting, it deals with an important, also socially, topic. The research problem, despite existing works on a similar topic, remains valid.

The methodology does not raise my doubts. The research was well planned, the research sample is large enough. The advantage of the work is to describe the research procedure in a precise and clear way.

In my opinion, although the work explored only one of many possible contexts, i.e. "readiness" to move, the work provides new and interesting information. The article broadens the knowledge on the mechanisms of adaptation to climate change. The work can also contribute to improving the adaptation solutions implemented in vulnerable areas, where relocation of the local community is an option.

In addition, the presented work may become an inspiration for other researchers. The significance of age or the experience of a climate disaster/hazard for readiness for relocation, identified in this paper, yielded varied results. This indicates the complexity of the problem and the need for continued research.

I recommend the work for publication.

Minor notes:

Line 36 - in “IPCC” = “I “ means The Intergovernmental not The International

Line 243 - second time Figure 2, wrong numbering

Table 3 - please add an explanation of the abbreviations used (under the table), e.g. JHS, SHS/Voc/Tech, GHC; what does "7.4ss" mean in the lowest row (3rd column)?

Lines 288-293 - please unify the notation of the % unit, it should be everywhere without spaces as in line 294

Figure 4 - no explanations for the symbols used in the figure (TA1-TA5; CA1-CA4, RP1-RP3)

Lines 353-354 - this small table should have a title and numbering

Table 5 - please add an explanation of the symbols in the table, such as χ2, p, Cramer's V

The list of references needs improvement in technical terms and adjustment to the editorial requirements of the journal.

Reviewer 2 Report

Adapting To Changing Climate: Understanding Coastal Rural Residents’ Relocation Intention In Response to Sea Level Rise

The paper gives an interesting account on adaptation to sea level rise in Ghana. The abstract gives a clear account of the topic and objectives but afterwards, a lot of focus on PMT has been given while the theory is already widely used and understood. Similarly, in the introduction, the theoretical framework mainly discuses the constructs of PMT that are easily understood by the readers so this section needs to be trimmed a bit. As the main aspect for which perceptions and adaptation mechanism is being investigated is the sea level rise but a little is noted about this fact in the introduction to effectively justify the selection of the issue as an exhibit of climate change. The selection of final respondents is not clearly described which needs the attention of the authors. The application of Exploratory Factor Analysis for the reduction of dimensions is well-suited and explained in sufficient detail. However, the use of ordered logit regression needs to be explained by giving its equation etc. The results on the logit regression in table 6 should highlight the significant factors while the discussion following it should have contextualized the findings with earlier studies, although there are some mentioning in Discussion section but it is not so explicit. The conclusion section is well put but it could also note some limitations of the work and future options for research, if any.

Reviewer 3 Report

Dear authors. I have attached the annotated PDF with several minor comments.

It is clear that you had problems writing your paper using the enumerated citation style. I have made several comments about this. Several journals use enumerated citations; therefore, you must master this writing style. The authors' names should be reduced to a few; your phrasing must allow inserting the citations within brackets (which must include all of them.......not in separate brackets). FOLLOW THE GUIDELINES!

This said, your paper is original and based on a strong statistical analysis. The results are interesting (less than the originality of the paper, however).

Yes, you should continue exploring the application of your methods to other adaptation strategies. I understand that the case was to explore relocation or not, test and learn from the results, which are good. Your conclusion about risk perception (against appraisal...from the literature) is relevant.

Finally: DO NEVER USE MITIGATE IN A CLIMATE CHANGE-RELATED PAPER WITH A MEANING DIFFERENT FROM CARBON EMISSION/SINK. In this paper, use reduce / risk reduction instead!

Author Response

Most of the comments from reviewer 3 were mostly grammatical and referencing style corrections. These have been rectified as stipulated in the PDF attached. Please see the revised manuscript for the corrections. Thank you.

Round 2

Reviewer 2 Report

Authors have sufficiently addressed my concerns but the comments on the giving some background and current status of sea level rise has not been fully addressed as well as trimming of the content on PMT. It would have been better if this could be addressed. Anyhow, the rest of the revision makes a lot of sense.

Author Response

Background and current status of sea level rise have been added from lines 35-47. Additionally, the content of PMT has been trimmed.